

# The signature of NAO and EA climate patterns on the vertical structure of the Canary Current Upwelling System

Tina Georg[1,2], Maria C. Neves[2,3], Paulo Relvas[2,4]

[1]now at: Institute of Geography, Johannes Gutenberg-Universität Mainz, Mainz, 55099, Germany
[2]Universidade do Algarve, FCT, Campus de Gambelas, Faro, 8005-139, Portugal
[3]Instituto Dom Luiz (IDL), Universidade de Lisboa, Lisboa, 1749-016, Portugal
[4]Universidade do Algarve, Center of Marine Sciences (CCMAR/FCT), Campus de Gambelas, Faro, 8005-139, Portugal

*Correspondence to*: Tina Georg (tina.georg@uni-mainz.de)

**Abstract.** The current study aims to analyse the vertical structure of the ocean during upwelling events using in situ and modelled data. Additionally, the influence of climate patterns, namely the North Atlantic Oscillation (NAO) and the East Atlantic (EA) pattern, on the vertical structure and their impact on the upwelling activity is assessed for a period of 25 years (1993–2017). The study focuses on the central part of the Canary Current (25–35° N) with persistent upwelling throughout the year with an annual cycle and strongest events from June to September.

Upwelling is determined using three different approaches: One index is calculated based on temperature differences between the coastal and the offshore area and two based on wind data and the resulting Ekman transport. Different data sets were chosen according to the indices.

Stable coastal upwelling can be observed in the study area for the analysed time span with differences throughout the latitudes and a time lag of four to five months between the wind and the temperature-based indices. A deepening of the isothermal layer depth and a cooling of temperatures is observed in the vertical structure of coastal waters representing a deeper mixing of the ocean and the rise of cooler, denser water towards the surface.

During years of a positive NAO, corresponding to a strengthening of the Azores High and the Iceland Low, stronger winds lead to an intensification of the upwelling activity, an enhanced mixing of the upper ocean and a deeper (shallower) isothermal layer along the coast (offshore). The opposite is observed in years of negative NAO. Both effects are enhanced in years with a coupled, opposite phase of the EA pattern and are mainly visible during winter months where the effect of both indices is the greatest. The study therefore suggests stronger upwelling activities in winters of positive North Atlantic Oscillation coupled with a negative East Atlantic pattern and emphasizes the importance of interactions between the climate patterns and upwelling.

## 1 Introduction

The upwelling regime of the Eastern Boundary Currents (EBCs) has a great impact on the primary production of the ocean and its ecosystem (i.e. Arístegui et al., 2009; Carr and Kearns, 2003; Gruber et al., 2011; Messié and Chavez, 2015; Santos et al., 2005; Troupin et al., 2010). As the Canary Current Upwelling System (CCUS) along with the California Current, the Humboldt Current and the Benguela Current cover about one percent of the ocean but make up for five percent of the primary production and more than 20 percent of the fish catch, the economic and scientific importance of these systems is evident (Bonino et al., 2019; Carr and Kearns, 2003; Chavez and Messié, 2009; Cropper et al., 2014).

Upwelling activity along the coastal area is dependent on the direction, strength and persistency of the winds, the coast as a solid boundary and the rotation of the Earth (Coriolis force). In the Canary Current, northeast trade winds lead to an Ekman transport away from the coast and upwelling of cooler and denser subsurface water in order to maintain mass balance (Ekman mechanism). Changes in the wind pattern with a changing climate will affect upwelling along the EBCs, however, a controversy exists on its extent. An intensification of the trade winds due to an increased on- to offshore pressure gradient



would lead to an intensification of upwelling (Bakun, 1990; McGregor et al., 2007; Polonsky and Serebrennikov, 2018; Siemer et al., 2021a). However, this intensification is not consensual since conflicting results based on the used data set were found (Narayan et al., 2010) as well as for the time and space of the upwelling activity along the Canary Current where the northern parts are influenced by the trade winds and the southern parts (below 20°N) by the southwest West African monsoon in summer

(Cropper et al., 2014). Conversely, several studies found decreasing upwelling intensity for the Canary Current with strong seasonal differences along with an increase in the SST (Barton et al., 2013; Bonino et al., 2019; Pardo et al., 2011). Studies on the net primary production which is closely linked to upwelling activity reveal stable to decreasing trends in the past years (Gómez-Letona et al., 2017; Siemer et al., 2021b). As studies have also found a weakening of the trade winds for the Canary Current and an intensification for other EBCs (IPCC, 2019; Sydeman et al., 2014), the impacts of climate change should be

assessed for each EBC separately.

In the Canary Current region, the wind patterns and, therefore, the upwelling activity along the coast, is mainly influenced by climate patterns like the North Atlantic Oscillation (NAO), the East Atlantic pattern (EA) and the Atlantic Multidecadal Oscillation (AMO). The NAO and its impact on upwelling and ocean dynamics has been investigated by several studies that showed mostly significant correlations (Benazzouz et al., 2014; Bonino et al., 2019; McGregor et al., 2007; Narayan et al.,

2010). In its biennial to decadal frequency of occurrence (Hurrell et al., 2001, 2003), the NAO is characterized by the pressure differences of the Icelandic Low and the Azores High which are well defined in the positive phase of the NAO (NAO+) and weak in its negative phase (NAO-). During years of NAO+, an enhancement of westerly winds, the trade winds and storms can be observed and during NAO- years a deceleration and a southerly shift of the westerly winds (Angell and Korshover, 1974; Luo et al., 2007; Visbeck et al., 2003). Additionally, a strengthening of the Atlantic Multidecadal Oscillation (AMOC)

and a deepening of the ocean's mixed layer have been detected in models for years of NAO+ (Delworth and Zeng, 2016; Yamamoto et al., 2020). The EA is a dipole shifted to the southeast with a low pressure system propagating from the north Atlantic to the western United Kingdom in its positive phase (EA+, Barnston and Livezey, 1987). Coupled, opposite phases of the NAO and the EA displaces the NAO centre of action towards the southwest causing alteration of the Azores High and more extreme weather conditions throughout Europe and the Atlantic Ocean (Bastos et al., 2016; Comas-Bru and McDermott,

2014; Häkkinen, 1999; Kalimeris et al., 2017; Yamamoto et al., 2020; Yan et al., 2004). Even though the AMO has been identified as a driver for upwelling, its low-frequency mode of 30–90 years makes it incompatible for the time span of the current study (Bonino et al., 2019; Schlesinger and Ramankutty, 1994; Yamamoto et al., 2020; Wang et al., 2017; Gallego et al., 2022), therefore, this study will focus on the impact of NAO and EA.

While the impact of climate patterns on the ocean surface and the weather across Europe has been subject to several studies,

little is known about their impact on the vertical structure of the upper ocean. In recent studies, extremes in variables such as precipitation, global land carbon sink or aquifer levels have been detected during periods of synchronisation amongst climate pattern (Cleverly et al., 2016; Bastos et al., 2016; Neves et al., 2019). Besides analysing the vertical structure and the ocean stratification during upwelling events, this study aims to identify the role of the climate patterns, especially the coupling between NAO and EA, in the CCUS.

**2 Data and Processing**

A flowchart summarizing the data sources and methods is shown in Fig. 1. All datasets have a time span of 25 years (1993–2017) and monthly sampling rate. Data on the vertical structure of the ocean comes from the Global Ocean Ensemble Physics Reanalysis dataset (PHY_001_026) obtained from the Global Reanalysis Ensemble Product (GREP), provided by the Copernicus Monitoring Environment Marine Service (2020: https://resources.marine.copernicus.eu). The data are provided as

a 3D grid with 1° x 1° horizontal resolution, divided into 75 depth levels from the surface down to a depth of 5,500 m, with vertical resolution varying from 10 m at 0–100 m depth, to 200 m at the ocean bottom. The variables considered are the



monthly-mean values of salinity and the seawater temperature (θ). In order to characterize upwelling, these variables are split into two regions defined as near- and offshore areas (Fig. 2). The nearshore area includes all grid cells between the coastline and the 200 m isobath (bounded by red lines), while the offshore area comprises all locations that are more than 500 km away

from the coastline (to the west of the 500 km offshore line). Locations around the Canary and Madeira Islands have been excluded to reduce their possible influence on the vertical structure of the ocean.

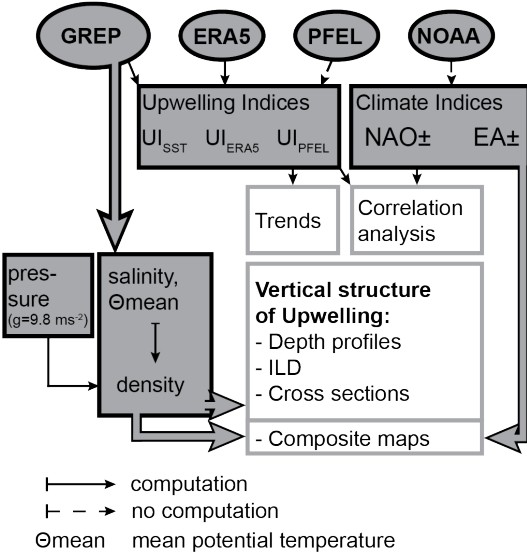

**Figure 1: Data sources and processing sequence**


+

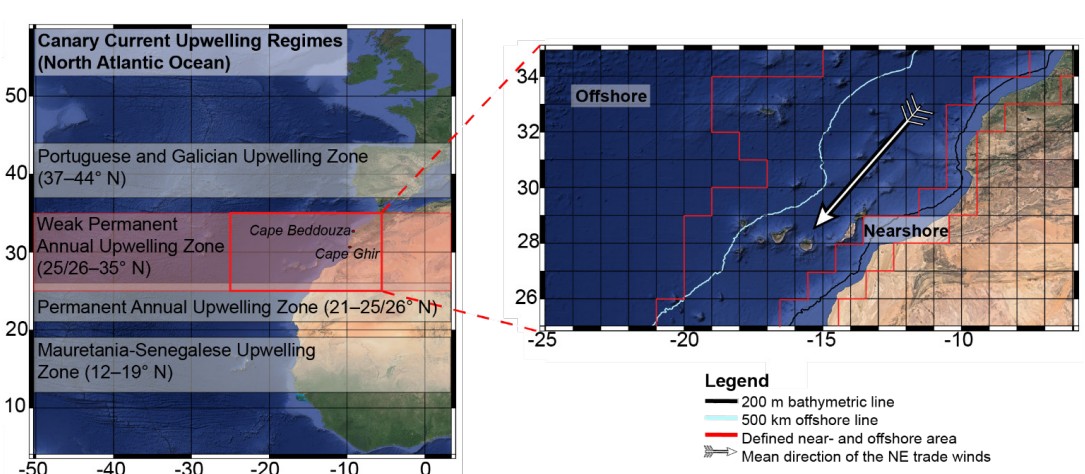

**Figure 2: Upwelling regimes in the Canary Current Upwelling Systems and the nearshore area of the study region (modified after Arístegui et al., 2009; Cropper et al., 2014, © Google Satellite 2022)**




The occurrence of upwelling is dependent on the direction and strength of the Ekman transport. The latter is a result of the prevailing wind conditions, the local bathymetry as well as the coastline geometry and the Coriolis force (i.e. Gómez-Gesteira et al., 2008; Cropper et al., 2014). Upwelling is determined using two different approaches: upwelling based on the Sea Surface Temperature ($UI_{SST}$) and on wind data ($UI_{ERA5}$).

The $UI_{SST}$ is computed using the mean temperature at the shallowest depth (0.50576 m) available in the GREP dataset, and is equal to the thermal difference between the offshore ($SST_{offshore}$) and coastal ($SST_{coast}$) areas, at a given latitude and time (Eq. 1):

$$UI_{SST} = SST_{offshore(lat,time)} - SST_{coast(lat,time)},\qquad(1)$$

The $SST_{coast}$ and $SST_{offshore}$ contain all grid cells in the near- and offshore areas, respectively. A positive $UI_{SST}$ indicates cooler temperatures near the coast. Upwelling events are recognized when the upwelling $UI_{SST}$ index exceeds the 5°C threshold. This is an upper bound to the thermal upwelling thresholds (varying between 2 and 5° C) used in other studies of the CCUS (Benazzouz et al., 2014; Cropper et al., 2014; Nykjær and Van Camp, 1994).

The source of data for the wind-driven upwelling index ($UI_{ERA5}$) is the ERA5 monthly averaged data on single levels, available from the Copernicus Climate Data Store (ECMWF Copernicus, 2019: https://cds.climate.copernicus.eu/). ERA5 is the fifth generation ECMWF reanalysis for the global climate and weather, combining modelled data with observations from across the world. The ERA5 variables required in this study are the horizontal components of the surface wind, available at 0.25° x 0.25° horizontal resolution. The calculation of $UI_{ERA5}$ is done in three steps (i.e. Bakun and Nelson, 1991; Cropper et al., 2014). Firstly, the zonal ($\tau_x$) and meridional ($\tau_x$) components of the wind stress are calculated using the wind speeds ($W_x$, $W_y$), the density of the air ($\rho_a$ = 1.22 kg m$^{-3}$), and a drag coefficient ($C_d$ = 1.14x10$^{-3}$, dimensionless, Large and Pond, 1981, Eq. 2):

$$\tau_x = \rho_a C_d (W_x^2 + W_y^2)^{\frac{1}{2}} W_x \text{ and } \tau_y = \rho_a C_d (W_x^2 + W_y^2)^{1/2} W_y,\qquad(2)$$

Secondly, the zonal (Qx) and meridional (Qy) components of the Ekman transport are calculated taking the density of seawater ($\rho sw$ = 1,025 kg m-3) and the Coriolis parameter (f = 2$\Omega$sin($\theta$) with $\Omega$ = 7.292x10-5 s-1 and $\theta$ = latitude) into consideration (Eq. 3):

$$Q_x = \frac{\tau_y}{\rho_{sw}f} \text{ and } Q_y = \frac{-\tau_x}{\rho_{sw}f},\qquad(3)$$

Finally, the $UI_{ERA5}$ is calculated using the Ekman transport and the coastline geometry ($\phi$ = mean angle between the shoreline and the equator, Eq. 4):

$$UI_{ERA5} = -\sin\left(\frac{\varphi-\pi}{2}\right) Q_x + \cos\left(\frac{\varphi-\pi}{2}\right) Q_y,\qquad(4)$$

For the $UI_{ERA5}$ index, the threshold for upwelling value was set to 1.5 based on the results.

For comparison and validation purposes, a third upwelling index ($UI_{PFEL}$) has been downloaded from the global monthly upwelling index database provided by the Pacific Fisheries Environmental Laboratory (PFEL) from the U.S. National Oceanic and Atmospheric Administration (NOAA). This index is based on the strength of the wind forcing and can be obtained for any point on any coastline from the PFEL's live access server (NOAA Fisheries, 2019: https://oceanview.pfeg.noaa.gov/services). The NAO and EA climate indices were retrieved from NOAA's Climate Prediction Center (2020a, 2020b: https://psl.noaa.gov/data/climateindices) at monthly temporal resolution and aggregated for the winter months, from December to March (DJFM). NOAA calculates these indices by rotated PCA (principal component analysis) of monthly means of the 500-mb geopotential height anomaly over the northern hemisphere. Positive and negative phases of the NAO and EA indices are defined by winter index values above 0.5 and below -0.5, respectively (Trigo et al., 2004).





The vertical structure of the ocean is inferred from variations of the seawater's density and temperature throughout depth.

Assuming a constant change of pressure with depth and the use of the gravitational acceleration of the Earth (g = 9.8 ms$^{-2}$), the density is computed from the salinity, pressure and mean temperature (GREP data) using the Gibbs Seawater (GSW) toolbox for the Thermodynamic Equation of Seawater 2012 (TEOS-10, McDougall and Barker, 2011). Subsequently, vertical profiles of density and temperature have been extracted and plotted from the GREP grid at times (months) corresponding to detected upwelling events. The extraction has been carried out for each latitude between 25.5° N and 34.5° N at 1° latitude

step, for the coast and offshore areas separately.

In the upper ocean changes in the kinetic energy from radiation, winds, waves, or horizontal advection by currents, among other factors, lead to mixing of the water up to a certain depth. The Isothermal Layer Depth (ILD) corresponds to the maximum depth of ocean mixing and, therefore, to the thickness of the physical properties (temperature) of the ocean being quasi-homogeneous. In this study the ILD is defined as the depth where the temperature differs by less than 0.8°C from the

temperature at the reference depth of 9.82 m (Chu and Fan, 2010 and 2011; Kara et al., 2000). The ILD has been computed for each upwelling event and its mean plotted in the vertical profiles. In addition, to examine the impact of climate patterns on the ocean stratification, the mean ILD has also been computed for positive and negative phases of the NAO and EA and for their neutral phases.

The upwelling indices have been correlated with each other and related to the coast and offshore mean ILD. Additionally,

these variables (UI's and ILD) were correlated with the climate patterns (NAO and EA). The relationships have been tested using the Pearson's correlation coefficient at a significance level of 0.05. The analysis of trends in the time series have been assessed using a linear regression model. Finally, composite maps have been created to explore the links between upwelling indices, the vertical structure of the ocean, and the climate patterns. The composite plots of upwelling are obtained by summing the plots at all times (months) corresponding to detected upwelling events. The same procedure is applied to obtain the

composite plots for the positive, negative and coupled phases of the climate indices.

### 3 Results and Discussion

### 3.1 Upwelling Indices: Trends and Correlations

The normalized upwelling indices have been plotted in a grouped boxplot showing its distribution per latitude (Fig. 3a). All UI's reveal uni-modal distributions with the $UI_{SST}$ data being spread widest and the $UI_{ERA5}$ and $UI_{PFEL}$ data slightly negatively

skewed. The highest $UI_{SST}$ occurs around 27.5° N, followed by decreasing values with increasing latitude. The other indices show a similar decreasing tendency with increasing latitude. When analyzed as a function of time, the time-series of the $UI_{SST}$, $UI_{ERA5}$ and $UI_{PFEL}$ reveal mostly positive values and a predominant occurrence of upwelling with a clear annual cycle (Fig. 3b). The highest values of the indices corresponding to the exceedance of the upwelling thresholds, occur in the summer months, especially in July and August. Besides the variation on the temporal scale, the strength of the upwelling events and

their spatial extent differs among indices. Overall, the results of the current study are in accordance with Narayan et al. (2010) and underline the issue of upwelling indices being very dependent on the used data sets and processing methods. They show the importance of studying upwelling using several methodologies and of adapting the calculation of indices to each geographical location, in order to make adequate predictions of their variability in the future.

There is a time lag of four to five months between the wind-based indices ($UI_{ERA5}$ and $UI_{PFEL}$) and the $UI_{SST}$ (Fig. 3b). Similar

time delays of the SST-based index have been found in previous studies, although with smaller lags (~ 2 months). Nykjær and Van Camp (1994) ascribed it to a possible influence of the bottom topography whereas Benazzouz et al. (2014) listed differences in the inertia of atmosphere and ocean or advection processes as possible causes.





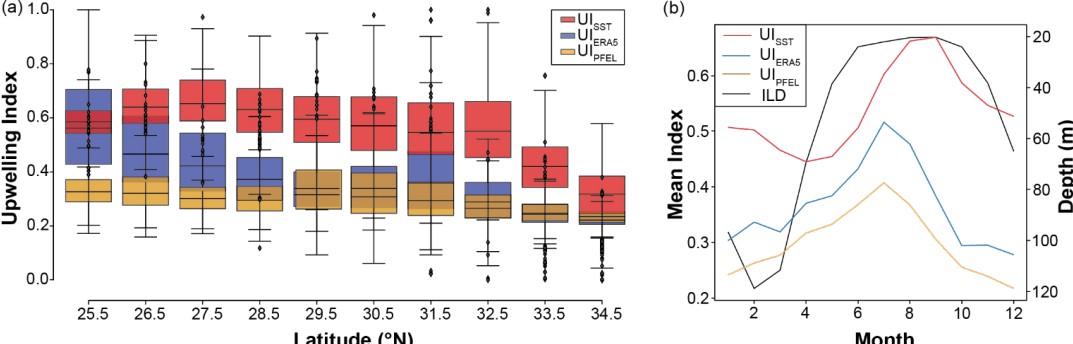

**Figure 3: Box-plot of normalized upwelling indices as a function of latitude (a) and monthly climatology of normalized upwelling indices and ILD averaged over the whole study region (b)**

Calculated trends are small and show mixed negative (-0.0008, $UI_{SST}$) and positive (0.1, $UI_{PFEL}$) tendencies in all upwelling indices, with differences depending on the latitudes. Therefore, no consistent trend has been detected in neither of the UI's during the selected time frame of 25 years. These results rather indicate a stable coastal upwelling during the past decades and give no support to previous contradictory findings of either an intensification or weakening of upwelling in the CCUS. Previous studies leading to the controversy (Bakun, 1990; Cropper et al., 2014; McGregor et al., 2007; Narayan et al., 2010; Polonsky and Serebrennikov, 2018) considered time spans of 30-60 years, hence, differing results can also be due to the temporal scale and the short time period used for this study should be used with caution when an assessment of long-term changes is done (Abrahams et al., 2021). Additionally, the usage of different data sets, defined study areas and thresholds makes a comparison less robust.

The correlations between the different upwelling indices are all significant with the highest degree of correlation found between $UI_{ERA5}$ and $UI_{PFEL}$ (Pearson's coefficient of correlation of 0.74). Comparatively, only moderate correlation was found between the wind and the SST-based indices ($UI_{SST}$ versus $UI_{ERA5}$: 0.548, $UI_{SST}$ versus $UI_{PFEL}$: 0.437). Average winter upwelling indices show moderate to good correlations with climate indices, especially wind-based upwelling indices and NAO. Good correlations between the winter NAO, wind intensity along the CCUS and upwelling had already been observed in previous studies (Cropper et al., 2014; Marrero-Betancort et al., 2020).

**3.2 Vertical Structure of Upwelling**

The vertical structure of the upper ocean during upwelling is the same regardless of the upwelling index used (wind-based or SST-based), so the vertical profiles and cross-sections are only shown for upwelling detected according to the $UI_{SST}$ threshold. Fig. 4 shows an illustrative example of vertical profiles corresponding to a strong upwelling event in September 1995. While single profiles at different latitudes are shown for the coastal area, the high number of offshore profiles have been enclosed in an envelope showing only the minimum, mean and maximum values at each depth step for the sake of clarity. As expected for upwelling, all vertical profiles reveal lower temperatures and higher density values in the coast than in the offshore area at the same depth. Differences between coast and offshore areas are maximum at the surface. Thus, the surface temperature of the upwelled water in the coast is around 20° C, whereas offshore the SST is around 25° C. The longitudinal thermal difference decreases with depth, but the nearshore temperatures are still below and the salinity above the offshore values up to a depth of 100-150 m. Below that depth, the differences in temperature and density between the near- and offshore areas vanish.

The ILD represents the limit of the sea-air-interaction at these scales and thus the maximum depth until which mixing influenced by kinetic and energy in the ocean occurs (Chu and Fan, 2011; Sprintall and Tomczak, 1992). In the study region the ILD shows a strong annual cycle (Fig. 3b) with shallower mean depths in summer (~20 m) than in winter (100-120 m). In





general, deeper ILDs during winter are explained by increased storm activity, stronger winds and greater heat losses at the surface as well as by negative buoyancy forces leading to more efficient mixing (Troupin et al., 2010; Yamaguchi and Suga, 2019). During summer, in contrast, high stratification is favoured due to greater surface warming through solar radiation (Barton et al., 2013). Still, when compared to normal summers we observe a deepening (of 1 to 2.5 m) of the ILD during upwelling events, especially nearshore. For example, during September 1995 the mean ILD reached 26.45 m at the coast and 24.48 m offshore (Fig. 4) while the average values for neutral summers was 22.2 and 20.4 m in the coast and offshore areas, respectively. Similar observations have been made in previous studies (Bessa et al., 2019, 2020). Nevertheless, the ILD alone is not an indicator of upwelling as along-shore wind and Ekman transport play the major role (Benazzouz et al., 2014; Polonsky and Serebrennikov, 2018).

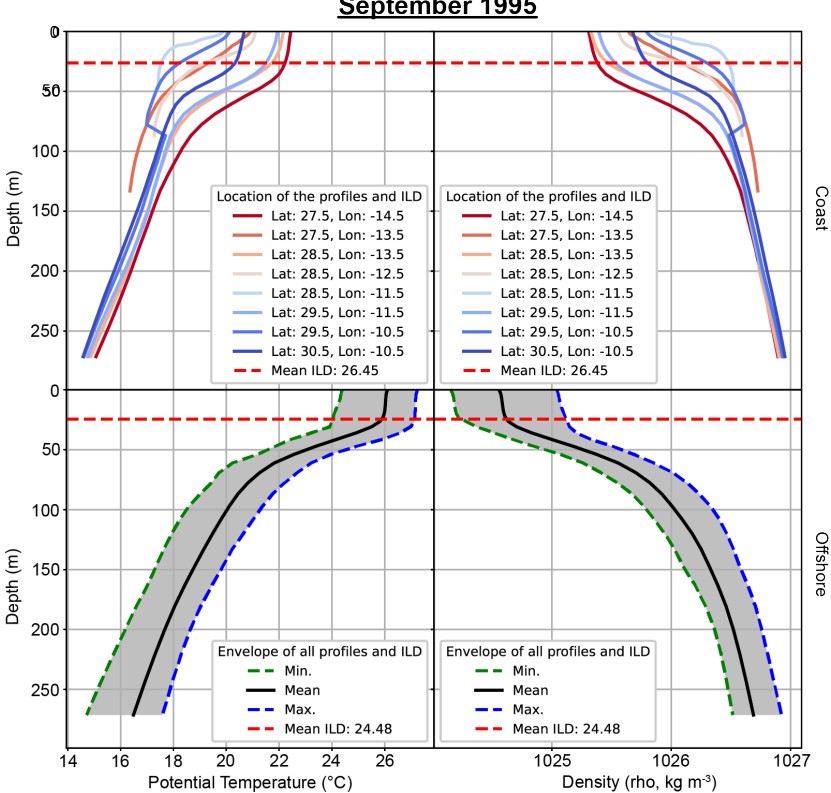

**Figure 4: Depth-profiles of density and temperature at different latitudes, at the coast and offshore. Example of an upwelling events detected in September 1995**

The cross-sections in Fig. 5 are examples of composite maps showing the temperature at two particular latitudes. The first corresponds to the sum of 3 upwelling events detected at 26.5°N while the second corresponds to the sum of 14 events detected at 30.5° N. Higher temperatures at the surface are clearly seen offshore to the west of -30°W while cooler temperatures near the coast indicate active upwelling. The cold-water plume with temperatures of less than 18°C ascends from depths of more than 200 m and distances of more than 1500 km from the coast. At the surface the vertical flow narrows and is focused at less than 100 km from the coastline, but the precision of its detection is limited by the resolution of the dataset. The isotherms are closer to each other at the surface in the offshore area with an increasing space between them with depth and towards the coast. The cooler temperatures along the continental shelf and the gradient of the isotherms sloping towards the surface clearly indicate the vertical flux of subsurface water towards the surface (Ekman suction) in order to replace the offshore Ekman

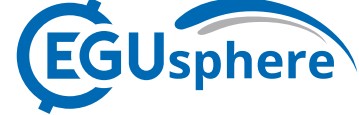

transport of surface water. With depth, the stratification increases and the differences between the near- and offshore area

decreases, as already observed in the depth-profiles (Fig. 4).

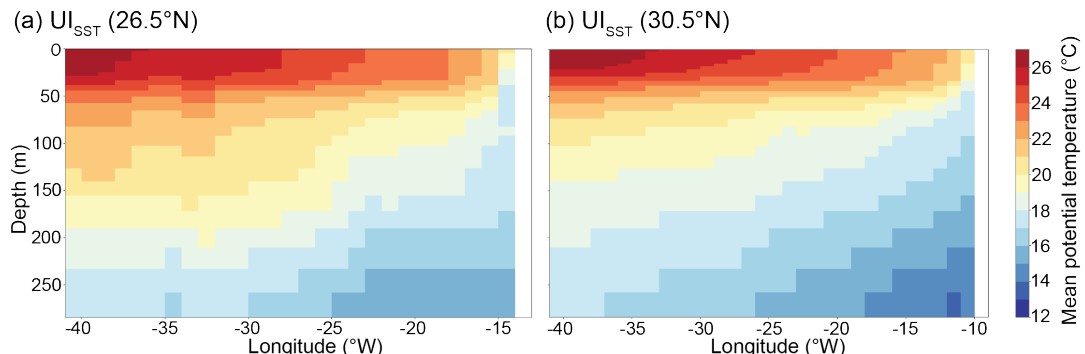

**Figure 5: Representative vertical cross-sections of temperature during upwelling at 26.5° N (a) and 30.5° N (b)**

Based on the climatology of the vertical water column velocity, derived from the Global Ocean Data Assimilation System
(GODAS) reanalysis, the region around 30° N has been identified as an area where upwelling weakens occasionally (Cropper
et al., 2014). However, the cross-section at 30.5° N displays slightly more intense upwelling than the cross-section at 26.5° N
(Fig. 5). In fact, these differences in the vertical structure of upwelling at different latitudes are not particularly significant and
reveal the spatio-temporal variability of the intensity of upwelling in the study area. That variability is mainly a consequence

of changes in the strength and meridional position of the trade winds but is also affected by the width and shape of the
continental shelf, as well as by local wind circulations related to the orography and land-ocean contrasts. Despite that
variability, the results of this study indicate stable coastal upwelling conditions extending from 26° N up to 32.5° N, which is
in agreement with other works (Arístegui et al., 2009; Cropper et al., 2014; Nykjær and Van Camp, 1994).

### 3.3 Signature of Climate Patterns on the Vertical Structure

The impact of NAO and EA on the vertical structure of the ocean is only visible in the winter, since their influence is supplanted
by the strengthening of trade winds during summer months. Horizontal composite maps of the temperature are presented for
the NAO phases, and for their couplings with opposite phases of the EA, at different depths (Fig. 6). Besides considering the
temperature at the surface (0.5 m), the depth steps in the upper ocean have been chosen according to the range of the ILD
observed in the winter months (Fig. 3b). The lowest depth displays the conditions where the influence of the atmosphere

through wind and heat flux is not present anymore.





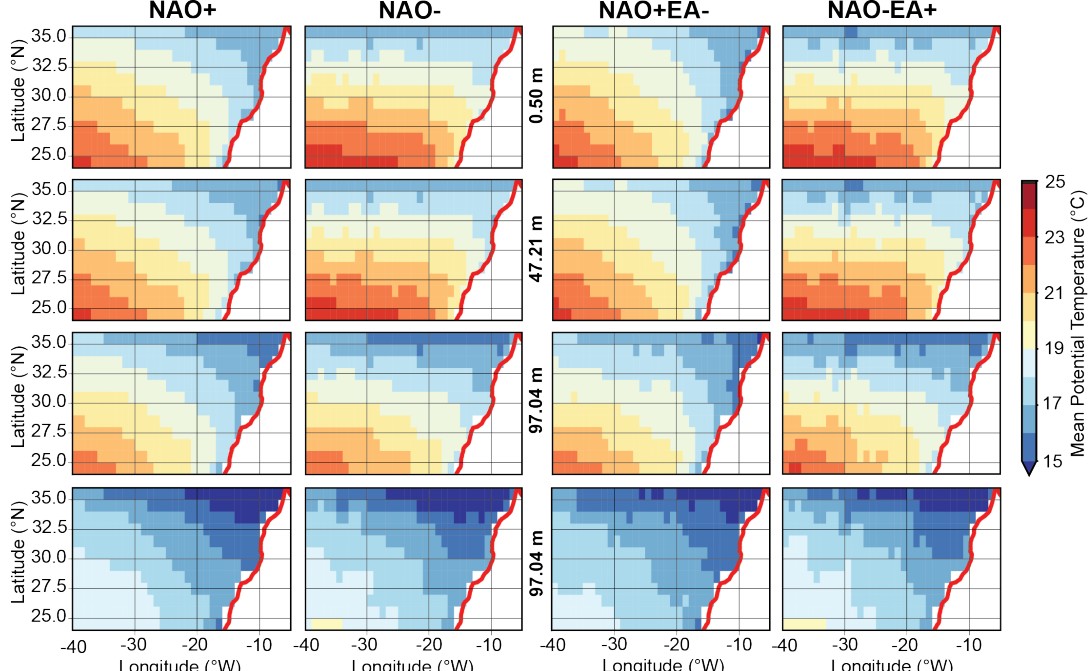

**Figure 6: Composite maps of temperature for positive and negative NAO and opposite NAO and EA phases showing the spatial extent of upwelling at different depths, red line represents the coastline**


There is an overall northeast to southwest gradient in the temperature, with colder waters in the NE and warmer waters in the SW corners of the study region (Fig. 6). In addition, one can observe a narrow strip of cold water (temperature < 18°C) adjacent to the coastline, which is indicative of upwelling. The NE–SW temperature gradient and upwelling zone are more evident during NAO+ and coupled NAO+EA- phases, particularly at the surface and at ~50 m depth. The temperature gradient persists at deeper levels (~ 100 and 200 m) but at these depths the patterns are nearly identical regardless of the climate phases. In contrast, the superficial patterns (depth ≤ 50 m) display a more meridional distribution during NAO- and NAO-EA+ phases, particularly in the offshore, with more reduced upwelling near the coastline. The observed patterns in the ocean temperature are a footprint of the large-scale atmospheric patterns during the winter. Thus, during NAO+ phases the strength of the Azores high-pressure system increases and generates stronger NE winds in the study region, the opposite occurring for NAO- phases. The stronger (weaker) winds for NAO+ (NAO-) can explain the intensified (lessen) upwelling and NE–SW temperature gradients at the ocean superficial levels.

Despite the moderate resolution of the dataset, it is also interesting to note that the intensity and extent of the upwelling strip along the coastline is greatest for coupled NAO+EA- events. The extreme upwelling signature of these events can be observed down to ~100 m depth. On the other hand, the persistence of the same temperature patterns at deeper levels (> 50 m), regardless of the climate phases, indicates higher inertia to changes and larger memory effects at depth and are consistent with the stable directions (NE-NNE) of the predominant winds in this region, despite the seasonality in their intensity (Marrero-Betancort et al., 2020).

The average ILD during winter months (DJFM) for different phases of the climate indices confirms the importance of NAO and EA couplings in the generation of extremes (Fig. 7). The ILD is greatly influenced by weather patterns and presents opposite variations at the coast and offshore areas. Variations are maximum near the coast (up to ~ 40 m) between a minimum of 75 m for NAO+EA- phases and a maximum of 155 m for NAO-EA+ phases. Deepening of the ILD near the coast with



NAO+EA- coupled phases is consistent with intensified alongshore NE winds, intensified Ekman transport, greater upwelling and therefore vertical mixing. The opposite shallowing of the ILD occurs for NAO-EA+ phases when there is a weakening of NE winds and very little signs of upwelling.


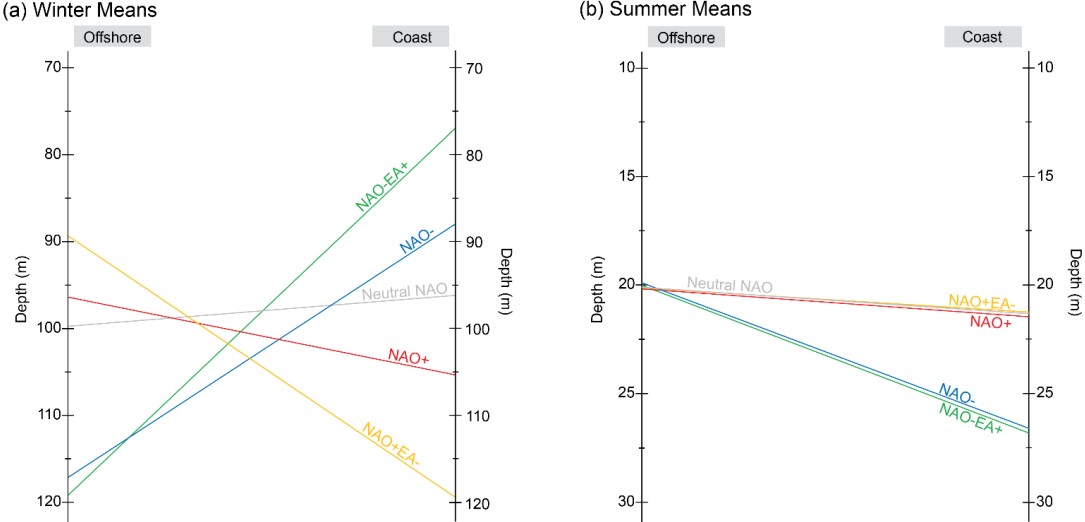

**Figure 7: Schematic representation of the mean ILD for NAO and EA phases during summer (a) and winter (b)**

The influence of NAO and EA on winter upwelling along the CCUS had already been recognized by Cropper et al. (2014)
among others (Benazzouz et al., 2014; Bonino et al., 2019; McGregor et al., 2007; Narayan et al., 2010). However, few studies so far have recognized the importance of interactions among NAO and EA, and only recently did we become aware that their couplings or superpositions, as well as the temporal shifts in their synchronization, may prompt extremes in several land variables such as net biome productivity, global land carbon sink or aquifer levels (Cleverly et al., 2016; Bastos et al., 2016; Neves et al., 2019). The present study further extends previous results on the role of NAO and EA in the study area, by showing
that coupled NAO+EA- phases also correspond to extremes in upwelling along the CCUS during winter.

**4 Conclusion**

The temporal and spatial extent of upwelling along the Canary Current Upwelling System was analysed from 1993–2015. Three different Upwelling Indices have been calculated, one based on the SST data ($UI_{SST}$) and two based on wind data ($UI_{ERA5}$ and $UI_{PFEL}$). Despite revealing upwelling in the selected area between 25 and 35°N, the results of the indices differ in their
strength and extent. Small negative or positive trends of the calculated indices imply stable coastal upwelling conditions in the past 25 years. Between the wind-based indices and the $UI_{SST}$, a time lag of four to five months was found which is greater than detected in previous studies along the Canary Current Upwelling System.

During detected upwelling events, the surface waters are cooler and denser at the coast in comparison to the offshore values resembling Ekman transport towards the offshore area and Ekman suction along the coast. This signature is represented by the
isotherms sloping towards the surface in the coastal area as shown in the cross-sections of the temperatures for upwelling events. The differences in temperature and salinity between the near- and offshore area decrease with depth. Ocean mixing and stratification was assessed through the calculation of the ILD. In dependence of the increased storm activity during the winter months and, therefore, an increased air-sea-interaction, the ILD deepens in winter and lowers in summer. An additional deepening of the coastal ILD was observed during upwelling events.



The changes of the ILD are striking when taking the climate patterns into account. The strengthening of the Azores high-pressure system during winter with NAO+ and the resulting stronger NE trade winds lead to enhanced mixing of the upper ocean in the coastal area. Thus, the ILD deepens along the coast, and gets shallower in the offshore area. The opposite can be observed during NAO- years and both occurrences are intensified during years of coupled, opposite phases of NAO and EA. The same impact of synchronised NAO and EA indices becomes visible in the horizontal composite maps (fig. 6). In years of
NAO+ there is a superficial (up to 50 m) temperature gradient from NE–SW and an evident upwelling zone. The latter extends deeper (up to 100 m depth) during NAO+EA- years. In contrast, during NAO- and NAO-EA+ a more meridional distribution can be observed at the surface and offshore, although the NE–SW temperature gradient in deeper levels is persistent regardless of the climate phases.

The study suggests that stronger upwelling along the CCUS in winter months is observed during coupled NAO+EA- phases.
It therefore emphasises the impact of coupled phases of climate pattern on extreme events in the ocean.

## Data Availability

The raw data used to create the plots in the paper are available through Copernicus Monitoring Environment Marine Service (GREP: https://resources.marine.copernicus.eu, data set: PHY_001_026), ECMWF Copernicus (ERA5: https://cds.climate.copernicus.eu), NOAA (EA: https://www.cpc.ncep.noaa.gov/data/teledoc/ea.shtml, NAO:
https://www.cpc.ncep.noaa.gov/products/precip/CWlink/pna/nao.shtml), and NOAA Fisheries (PFEL: https://oceanview.pfeg.noaa.gov/products/upwelling/bakun).

## Author contributions

MN and PR developed the research question, supervised the work and reviewed the content, TG conducted the data processing and prepared the manuscript. All authors contributed to the interpretation and discussion of the data.

## Competing interests

The authors declare that they have no conflict of interest.

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
