# Peer review of "The signature of NAO and EA climate patterns on the vertical structure of the Canary Current Upwelling System"

_EGUsphere, 2022_

## Author Comment (AC1)

**The signature of NAO and EA climate patterns on the vertical structure of the Canary Current Upwelling System**

Tina Georg, Maria C. Neves, Paulo Relvas

**Response to Referee #1 (RC1):**

**Referee:** The manuscript describes analyses of ocean and other model output in the vicinity of the Canary Current Upwelling System. Specifically, it examines three different metrics of upwelling system indices, the vertical temperature structure of nearshore and offshore water columns, and the patterns of temperature that are correlated with wintertime indices of the North Atlantic Oscillation and East Atlantic atmospheric patterns. Oceanic temperature is obtained from the Global Ocean Ensemble Physics Reanalysis dataset obtained from CMEMS.

The authors find that (1) different upwelling indices have different values and seasonal cycles, (2) the isothermal depth of nearshore profiles during upwelling is less than that for offshore profiles, and (3) upwelling is most intense during the positive phase of the NAO and (4) especially that in combination with the negative phase of the EA.

In my opinion, the main advance of this paper is item (4) and this result is interesting and useful. Analysis of the upwelling indices appear and vertical structure are to me less novel, though it might be argued that they raise interesting questions (e.g., about what is the best upwelling index to use) and provide useful context (e.g., typical and anomalous isothermal layer depths) for the remainder of the paper.

**Response:** We are very grateful for your detailed review and suggestions for improvement and thank you for your effort. Please find below the reply to your comments.

**Main Recommendation:**

**Referee:** I think the manuscript would benefit from the authors choosing the best upwelling index to characterize the upwelling, presenting only that, and then expand on the NAO/EA parts of the paper. Is the comparison of UI important? Is the 5 month time-lag between UI_ERA and UI_SST important beyond showing that indices based on different data and approaches are different?

**Response:** Although the usage and comparison of different upwelling indices is not new, we used different data sets to detect upwelling as precisely as possible. Additionally, since we compare the vertical structure of upwelling to the climate patterns, our aim was to include a wind- and a temperature-based UI. For that reason, in the new version of the manuscript we include UI_PFEL and UI_SST, covering both the surface wind that changes with the phases of NAO/EA and the temperature which is important for the assessment of the vertical structure of the ocean. We agree that the usage of two different wind-based indices does not add any new insights to the current study, so we decided to remove the UI_ERA5 and keep the UI_PFEL as also suggested by other reviewers.

**Referee:** It would be helpful if the authors would include a local map of winds associated with the NAO and EA patterns (to understand their impact on local upwelling), along with their time-series, showing highlighted periods of upwelling that were used for averaging the model. At the moment, the method by which averaging is done is not clear. How many days contributed to the NOA+, NAO-, NAO+EA, and NAO-EA+ fields shown in Figure 6 and 7. How much uncertainty is there in the averages calculated? Perhaps time-series or pdfs of upwelling events associated with different climate conditions could further support the argument that statistics change in different climate conditions. How much uncertainty is there in the averages calculated?

**Response:** We agree with the reviewer and added the following figures and table to the manuscript:

i)   A map with the wind fields (ERA5 data) during NAO+/- and EA+/- phases (Fig. 2)

[Figure]

**Figure 2: Wind conditions and SST during positive and negative NAO, and during combined opposite phases of NAO and EA. Wind data obtained from ERA5, SST data from Copernicus (for detailed information on NAO+, NAO-, NAO+EA-, and NAO-EA+ see tab. 1 and fig. 4)**

ii)   A table with years of NAO and EA positive and negative phases (Table 1).

| NAO | 1993 | 1994 | 1995 | 1996 | 1999 | 2000 | 2007 | 2008 | 2010 | 2012 | 2014 | 2015 | 2016 | 2017 |
|-----|------|------|------|------|------|------|------|------|------|------|------|------|------|------|
| EA | 1993 | 1994 | 1995 | 1996 | 1999 | 2000 | 2007 | 2008 | 2010 | 2012 | 2014 | 2015 | 2016 | 2017 |

**Table 1: Phases of NAO+ (red) and NAO- (blue) with the corresponding phases of the EA. Combined opposite phases are marked in yellow.**

iii)   A figure of the climatology of NAO and EA along with the used threshold (0.5) and the combined opposite phases (Fig. 4).

[Figure]

**Figure 4: Winter means of NAO and EA. Positive and negative phases are marked with + and -, respectively, combined opposite phases are marked in yellow.**

iv) Plot of the upwelling indices for different climate conditions

[Figure]

**Figure 5: Box-plot of normalized upwelling indices as a function of latitude (a), monthly climatologies of normalized upwelling indices from 1993-2017 (b), during years of NAO+ (c), and during years of NAO- (d)**

We also agree that the method of averaging was not clear enough. We added the following sentence to the manuscript which hopefully answers the question (Section 2, from line 169):

The composite plots of upwelling are obtained by summing and averaging the temperature of each grid cell for all months with a positive and negative NAO as well as months with combined opposite phases of NAO and EA (average computed for the winter months (DJFM) of the years listed in Table 1).

The new plots of the upwelling indices for different climatic conditions (NAO+, NAO-, NAO+EA- and NAO-EA+) shown in Fig. 5 provide further support to our arguments.

The following was added to the text (section 3.1, from line 198):

The impact of the climate patterns on the upwelling indices is well illustrated in Fig. 5 which displays $UI_{SST}$ (5a) and $UI_{PFEL}$ (5b) averaged over years of positive and negative phases (e.g. $UI_{SST}$ for NAO+ correspond to monthly values averaged over NAO+ years). Averages over all the years (1993-2007) as well as averages over years of coupled phases (marked in yellow in Table 1) are also shown for comparison. More differentiated and consistent relationships can be inferred for $UI_{SST}$ than for $UI_{PFEL}$, particularly from December to July (Fig. 5a). Larger values of $UI_{SST}$ are associated with positive phases of the winter NAO, as expected due to the strengthening of the alongshore winds during NAO+ (Fig. 2). However, Fig. 5 also shows that maximum upwelling indices occur in years of NAO+EA- combinations, while minimum values occur during either NAO- years or NAO-EA+ combinations. Additionally, the time lag between the $UI_{SST}$ and $UI_{PFEL}$ differs with the phases of the climate pattern. Previous studies identified a lag of 2 month which is only visible during years of NAO+EA-. During the other phases and throughout the whole period of the study, a greater time lag of 3 months becomes visible. This suggests that besides the influence of the bottom topography (Nykjær and Van Camp, 1994, *J. Geophys. Res.*, **99**, 14197-14207), differences in the inertia of atmosphere and ocean and advection

processes (Benazzouz et al., 2014, *Cont. Shelf Res.*, **81**, 38-54), the climate patterns and especially the combined phase of NAO+EA- play a role. Due to the limited number of years (only two) in which coupled NAO+EA- conditions occur (Table 1) caution is required in generalizing this result. Nonetheless, these results are qualitatively similar to a recent analysis of the impact of opposing NAO and EA phases during winter on precipitation, groundwater levels, and wind power generation in Portugal (Neves et al., 2019, *J. Hydrol.*, **568,** 1105-1117, and 2021, *J. Clean. Prod.*, **320**, 128828).

**Other comments**

**Referee:** Lines 115-124: The calculation of UI_ERA5 I think should rotate the winds to the alongshore direction and then calculate the wind stress, rather than the reverse as is done presently.
**Response:** Thank you for the comment. Since we decided to remove the UI_ERA5, this is not applicable in the current manuscript.

**Referee:** The results section (around lines 210), the authors claim that a change in the ILD of 1-2 m during upwelling events. There is no error analysis to show significance of this, but even if there was, is a 10% deepening important? And (as they point out), this is not a new result (Line 213). This section might be dropped.
**Response:** Indeed 10% is not much. Since we use the ILD during NAO/EA years as an indicator of changes in the upwelling activity and ocean stratification, we think it is important to mention the changes detected in the ILD in years where NAO/EA play a role. We did, however, reduce the section and re-write the paragraph at the end of section 3.2 as:

The ILD represents the limit of the sea-air-interaction at these scales and thus the maximum depth until which mixing influenced by kinetic and energy in the ocean occurs (Chu and Fan, 2011, *Oceans and Land Surface*, 1001-1008; Sprintall and Tomczak, 1992, *J. Geophys. Res.*, **97**, 7305-7316). In the study region the ILD shows a strong annual cycle with shallower mean depths in summer (~20 m) than in winter (100-120 m). In general, deeper ILDs during winter are explained by increased storm activity, stronger winds and greater heat losses at the surface as well as by negative buoyancy forces leading to more efficient mixing (Troupin et al., 2010, *J. Mar. Syst.*, **80**, 172-183; Yamaguchi and Suga, 2019, *J. Geophys. Res*, **124**, 8933-8948). During summer, in contrast, high stratification is favoured due to greater surface warming through solar radiation and the ILD deepens during years (Barton et al., 2013, *Prog. Oceanogr.,* **116**, 167-178). Still, when compared to normal summers we observe a deepening (of 1 to 2.5 m) of the ILD during upwelling events, especially nearshore. Nevertheless, the ILD alone is not a sole indicator of upwelling as along-shore wind and Ekman transport play the major role (Benazzouz et al., 2014, *Cont. Shelf Res.*, **81**, 38-54; Polonsky and Serebrennikov, 2018, *Izv. - Atmos. Ocean. Phys.*, **54**, 1062-1067).

**Referee:** Similarly, What's to be interpreted as important in Figures 4 and 5. They do show differences in coastal and offshore profiles, but the figures seems routine. Why characterize the vertical profiles or representative sections? As the authors point out, the description that they give are in agreement with other works (Line 243).
**Response:** We agree that the figure does not add any new finding to the study. We will remove it from the manuscript and would like to make it available as supplementary material. Although it is in agreement with other works, to our knowledge, the representation with those data and methodologies were never presented. Therefore, we would like to keep figure 5 (now fig. 6) in the manuscript to show the vertical structure of the ocean during upwelling and to make it easier to understand figure 6 (composite SST for the climate patterns, now fig. 7).

**Referee:** Line 178: The result that the trends in UI are small over 25 years is interesting and useful.
**Response:** Thank you for the comment.

**Referee:** Line 271: Minor comment: I think the authors mean "observed down to ~50 m depth"? Also Fig 6 has an error in listing 97.04 m twice. I think the authors mean ~200 m in the bottom row?

**Response:** There are still minor differences at a depth of ~100 m, however, we agree that the extreme signatures can be observed to a depth of ~50 m so we changed the depth in the manuscript. The bottom row represents the depth of 199.79 m, we will correct the error in the figure. Thank you for noticing.

**Referee:** Figure 7 is very interesting and compelling.
**Response:** Thank you very much for the comment.

---

## Author Comment (AC2)

**The signature of NAO and EA climate patterns on the vertical structure of the Canary Current Upwelling System**

Tina Georg, Maria C. Neves, Paulo Relvas

**Response to Referee #2 (RC2):**

**Referee:** The authors describe an analysis of upwelling off the NW African coast associated with the Canary Current system. A 25 year period is considered using data from various sources. A particular focus of this study is the relationship between upwelling and variability that may be driven by the North Atlantic Oscillation (NAO) and the East Atlantic (EA) teleconnection patterns. A new aspect of this study appears to be an analysis of the vertical structure of the water column associated with upwelling variability.

On the whole, the paper is well written and contains copious references to previous relevant literature. There are a few issues though that I would like to raise. I believe that this paper contains results that would be of interest to the community and once the issues mentioned below have been addressed, it should be appropriate for publication.

**Response:** Thank you for your review and the suggestions. We are very grateful for your effort.

**General comments:**

**Referee: (1)** For this reviewer, one of the major weaknesses of this study is the very coarse horizontal resolution of the GREP ocean data set that is used for analysis of SST and vertical structure. The horizontal grid spacing of GREP is only 1 degree which means that in reality the effective resolution is probably more like 3 or 4 degrees. On the otherhand, the width of the upwelling region due to Ekman divergence at the coast may only be ~ the Rossby radius of deformation, which is probably ~40km for the 1st barocinic mode. This is much shorter than the resolution of the GREP data. If wind stress curl is an important factor in enhancing coastal upwelling in the region, the width of the upwelling zone may be larger, but again still less than the GREP resolution. Therefore, this study is really more a reflection of how upwelling varies in the model ensemble described by GREP rather than in nature. While coastal upwelling is clearly being captured by GREP, it is undoubtedly a highly distorted view compared to the real world. This should be discussed clearly in the manuscript, at the outset and in the conclusions.

**Response:** Thank you for the comment. We are aware of the limitations of the GREP data. The aim of the study was to solely use in-situ measurements available from the World Ocean Atlas (WOA18). However, after analyzing the available data, we realized that we could not use only in-situ data for the scope of our study:

The WOA18 provides in-situ data from 1969 to 2019. The figure attached shows the available data in the defined coastal and offshore area for the whole time span (yellow dots). Only the red dots indicate data with the same latitude and the timing. With this data, it was not possible to obtain a good insight into (a) upwelling in general, (b) the changes with the phases of the climate patterns, and (c) changes over time could not be assessed. We, therefore, decided to use the GREP data as the model data are validated with field data, if applicable.

[Figure]

**Legend**
○ All available data (near- and offshore)
● Usable data to detect upwelling (same time and latitude)
— 200m bathymetric line
— 500km offshore line

**Available data of the WOA18. Please note: We will not add this figure to the manuscript**

To clarify the data (model vs. real world), we added the following to the text:

In section 2 (from line 77):
Since the data of the available in-situ measurements for the study area is scarce, we used data on the vertical structure of the ocean from the Global Ocean Ensemble Physics Reanalysis dataset (PHY_001_026) obtained from the Global Reanalysis Ensemble Product (GREP), provided by the Copernicus Monitoring Environment Marine Service (2020: https://resources.marine.copernicus.eu) to analyse the vertical structure of the ocean. The GREP data is produced using a numerical model (NEMO model on ORCA025 grid, Bernard et al., 2006, *Ocean Dyn.*, **56**, 543-567) with a surface forcing by ERA interim and data assimilation using satellite and in-situ data.

In section 4 (from line 354):
It is, however, necessary consider that the used dataset consists of modelled data. Even though the results are validated with in-situ measurements and satellite data, the model might not reflect the in-situ conditions of the ocean structure.

**Refereee: (2)** Two different upwelling indices (UIs) based on the wind were used: one is the standard PFEL product while the other is one that the authors compute based on equation (5). The definition of the PFEL index is not given in the manuscript, so it is not clearly what the relationship is between $UI_{PFEL}$ and $UI_{ERA5}$. Figure 3b shows that they vary consistently over time, so why consider both? Why not just use the accepted $UI_{PFEL}$? This needs further discussion and justification.
**Response:** As other reviewers suggested, we removed one of the wind-based UIs, namely the UI_ERA5. The aim of the different UIs was to compare the two different approaches and to verify our calculations. We agree that the comparison does not contribute to the main purpose of the study (that is the signature of climate pattern).

**Refereee: (3)** Figure 3b shows that there is a lag in the SST response and the upwelling indices. This is mentioned in the manuscript, and has been noted by others, but this manuscript sheds no further light on this issue. Studies of coastal upwelling by Marchesiello and Estrade (2010, *J. Mar. Res.*, **68**, 37-62) and Jacox et al. (2014, *GRL*, **41**, 3189-3196) have shown that coastal upwelling can be suppressed by onshore geostrophic flow leading to considerably less upwelling than might be expected based on the wind alone. I wonder if this might be the reason why the upwelling peaks later than the wind-based upwelling indices in the Canary Current system. This could perhaps be easily checked using GREP which presumably also contains the near-surface ocean current data.

**Response:** The use of "upwelling indices" is always problematic and controversial. They show strong limitations since they rely only on the forcing factor (the wind) or in the surface evidence (SST patterns). Upwelling is a subsurface process, forced at the surface, that occurs in the water column. Anyway, the vertical structure of the ocean does not take part in the definition of any widely used "upwelling index".

In this region, the CCUS, the possible distortion of the wind induced upwelling by onshore geostrophic flow must be attributed to the North Atlantic meridional density gradient. This is the forcing that can generate onshore (eastward) geostrophic flow, because of the sea level decline towards the pole that it induces. It is part of the JEBAR mechanism proposed by Huthnance (1984, *J. Phys. Oceanogr.*, **14**, 795–810). At the end, the onshore geostrophic flow will force alongshore poleward flows over the continental slope. Along with alongshore pressure gradients that are known to occur, they enforce three dimensionality to the upwelling process.

Of course, all these dynamics will affect the discrepancy between UIs. Other features, such as wind stress curl, bottom topography, or stratification variability, will also affect differently UIs based on the forcing (wind stress), or in the final surface evidence (SST). Marchesiello and Estrade (2010, *J. Mar. Res.*, **68**, 37-62) and Jacox et al. (2014, *GRL*, **41**, 3189-3196), along with Jacox et al. (2018, *JGR-Oceans*, **123**, 2018JC014187) mention these factors too. All these factors make the definition of "upwelling intensity" very broad.

The discussion of the physics that is behind the construction of the upwelling indices is out of the scope of the present research. Since our goal is to study long time-series and the subsurface response based on reanalysis, not on pure modelling efforts, we must make use of universal indexes, computed along decades through standardized methods.

Following your suggestion, that we kindly acknowledge, we have extracted and plotted the monthly climatologies of the eastward seawater velocity obtained from GREP (uo_mean). The resulting plot only shows negative values corresponding to a westward flow which is offshore. Therefore, the flow is in close agreement with the expected upwelling pattern, offshore Ekman transport in the upper layers all year, increasng during summer months. It seems that the onshore geostrophic flow is not enough to invert the cross-shore flow.

[Figure]

**Plot of the eastward seawater velocity (uo_mean) obtained from GREP. Please note: We will not add this figure to the manuscript**

The following sentence was added to the text (section 3.1, from line 285):
Additionally, other factors such as wind stress curl, stratification and onshore geostrophic flow may contribute to the time lag, as they interfere with the upwelling process and condition the upper ocean response to the wind forcing (Marchesiello and Estrade, 2010, *J. Mar. Res.*, **68**, 37-62).

**Specific comments:**

**Referee: (1)** Caption for figure 1: More information is needed here - all the acronyms and symbols should to be defined in the caption.
**Response:** We agree and adapted the caption as follows:

Figure 1: Flow chart of the Data sources for the vertical structure of the ocean (GREP: Global Reanalysis Ensemble Product data by Copernicus), the Upwelling Indices (UI, PFEL: Pacific Fisheries Environmental Laboratory by NOAA), the climate patterns (NOAA: National Oceanic and Atmospheric Administration, NAO: North Atlantic Oscillation, EA: East Atlantic pattern) and the processing sequence (ILD: Isothermal Depth Layer).

**Referee: (2)** Line 115 and equations (2) and (3): Non-standard notation is used here. I would suggest using u and v instead of $W_x$ and $W_y$, and U and V instead of $Q_x$ and $Q_y$.
**Response:** This is a good remark, thank you. Since we decided to remove the UI_ERA5, it is not applicable anymore.

**Referee: (3)** Equation (3): Can you comment on the relative role of Ekman divergence at the coast and wind stress curl? It is likely that both contribute in a significant way to upwelling, as they do at other upwelling centres.
**Response:** This is an interesting comment that we acknowledge. This is a question that has been focused in several investigations. We believe that wind stress curl and Ekman divergence are not independent. The relative role of both is highly dependent on the coastline configuration. We are certain that the wind curl will strongly modulate the offshore extent of the cold upwelled water, thus affect the SST based UI.

Anyway, since we remove the UI_ERA5, we will also remove the equations, and this discussion is now out of sense in this MS.

**Referee: (4)** Line 124: The upwelling threshold of 1.5 for UI$_{ERA5}$ needs some explanation/justification.
**Response:** We chose the threshold based on the results obtained from our calculations. It is a compromise between having too many data and being representative. As stated before, we removed the UI_ERA5, therefore, we will not add the explanation to the text.

**Referee: (5)** Line 127: Please provide a full definition of the PFEL upwelling here (see comment above).
**Response:** The PFEL index uses the x(EW)- and y(NS)-components of Ekman transport with the rotation angle; Ekman transport data can be obtained by location.

The following was added/changed in the text (section 2, 137):
The wind-driven upwelling index (UIPFEL) data has been downloaded from the global monthly upwelling index database provided by the Pacific Fisheries Environmental Laboratory (PFEL) from the U.S. National Oceanic and Atmospheric Administration (NOAA). This index is based on the strength of the wind forcing and can be obtained for any point on any coastline from the PFEL's live access server (NOAA Fisheries, 2019: https://oceanview.pfeg.noaa.gov/services). The data set provides the results for the zonal and meridional component of the Ekman transport (Bakun, 1990, *Science*, **247**, 198-201; Bakun and Nelson, 1991, *J. Phys. Oceanogr.*, **21**, 1815-1834) and allows the computation of the Upwelling Index in consideration of the prevailing coastline geometry using a function for Python which is also provided by NOAA.

**Referee: (6)** Caption for Fig. 3: Please explain the format of the boxplots in panel (a) (i.e. what do the coloured boxes and various tick marks represent?).
**Response:** The boxplots represent the distribution of the data showing outliers (dots on both sides), whiskers, quartiles and the median. The colors represent the different Upwelling Indices. After removing the UI_ERA5, the plot should be clearer.

**Referee: (7)** Line 178: What are you referring to here by "trend" - there are no plots presented that indicate a trend.
**Response:** The trend was calculated for each latitude in each UI using linear regression (see line 153, section 2), only the mean of the latitudes was added to the study (line 180). Throughout the latitudes, only a small variation is visible.

**Referee: (8)** Lines 187-189: Have you computed lagged correlation coefficients? What about the potential role of onshore geostrophic flow (see comment above)?
**Response:** Please see the response to (3) of general comments.

**Referee: (9)** Lines 210-214: The changes in depth for the ILD discussed here seem very small. Can you discuss their significance? The very low horizontal resolution of the GREP model data sets used must be an important limiting factor here.
**Response:** Other reviewers mentioned the same. We agree that the changes seem small on a local scale, however, when dealing with climatic impact a change of 10% is important.

**Referee: (10)** Figure 4: It would help to show the location of these profiles in Fig. 2. Also, this figure is not easy to read. Is there a better way of demonstrating how T and ρ vary with distance from the coast?
**Response:** Based on the reviews, we decided to remove the Figure 4 from the manuscript and make it available as supplementary material.

**Referee: (11)** Caption for Fig. 5: I don't think you mean "representative" here. According to the main text these are the average temperature profiles based on several events that exceed a threshold based on $UI_{SST}$ - is that correct?
**Response:** That is correct, thanks. We have removed "representative".

**Referee: (12)** Line 247: I think that you mean "combination" rather than "coupling." The reason why you chose to consider these particular combinations of the NAO (+ -) and EA (+ -) in Fig. 6 should to be explained.
**Response:** We changed "coupling" to "combination". The combination is used since previous studies (i.e. Bastos et al. 2016, *Nat. Commun.*, **7**, 1-9) have shown the impact of it on the weather conditions in Europe and the Atlantic Ocean. This is mentioned in the introduction but we agree to mention it again in the discussion to emphasize the importance of the combination.

**Referee: (13)** Line 250: Replace "not present anymore" with "largely absent"
**Response:** We replaced "not present anymore" with "largely absent".

**Referee: (14)** Figure 6: Please indicate more clearly the depths represented by each row of plots.
**Response:** We changed the position of the depths to the first plot of each row.

**Referee: (15)** Line 268: Replace "coupled" with "combined"
**Response:** We replaced "coupled" with "combined".

**Referee: (16)** Line 272: Rephase "...NAO and EA couplings..." as "...the combined influence of the NAO and EA in the ..."
**Response:** We changed "...NAO and EA couplings..." to "...the combined influence of the NAO and EA in the ...".

**Referee: (17)** Lines 296-297: Why is the lag larger in your study? See comment above about possible role of onshore geostrophic flow.
**Response:** See main comments (3).

**Referee: (18)** Section 4: It would be useful to discuss clearly what this study adds to the existing literature.
**Response:** We changed the conclusion as follows:
The temporal and spatial extent of upwelling along the Canary Current Upwelling System was analysed from 1993–2017. Two different Upwelling Indices have been calculated, one based on the SST data (UI_SST) and one based on wind data (UI_PFEL). Despite revealing upwelling in the selected area between 25 and 35°N, the results of the indices differ in their strength and extent. Small negative or positive trends of the calculated indices imply stable coastal upwelling conditions in the past 25 years. Between the UI_PFEL and the UI_SST, a time lag was found for the maximum values which changes with the phase of the climate patterns. Previous studies along the Canary Current Upwelling System identified a time lag of 2 months which only becomes apparent during years of NAO+EA-. In years of the remaining phases of the climate patterns and the overall period of the study a lag of 3 months was observed which emphasizes the role of the climate patterns on Upwelling. Additionally, highest values of both indices are linked to NAO+EA-.

During detected upwelling events, the surface waters are cooler and denser at the coast in comparison to the offshore values resembling Ekman transport towards the offshore area and Ekman suction along the coast. This signature is represented by the isotherms sloping towards the surface in the coastal area as shown in the cross-sections of the temperatures for upwelling events. The differences in temperature between the near- and offshore area decrease with depth. Ocean mixing and stratification was assessed through the calculation of the ILD. In dependence of the increased storm activity during the winter months and, therefore, an increased air-sea-interaction, the ILD deepens in winter and lowers in summer. An additional deepening of the coastal ILD was observed during upwelling events.

The changes of the ILD are striking when taking the climate patterns into account. The strengthening of the Azores high-pressure system during winter with NAO+ and the resulting stronger NE trade winds lead to enhanced mixing of the upper ocean in the coastal area. Thus, the ILD deepens along the coast, and gets shallower in the offshore area. The opposite can be observed during NAO- years and both occurrences are intensified during years of coupled, opposite phases of NAO and EA. The same impact of synchronised NAO and EA indices becomes visible in the horizontal composite maps (fig. 7). In years of NAO+ there is a superficial (up to 50 m) temperature gradient from NE–SW and an evident upwelling zone. The latter extends deeper (up to 100 m depth) during NAO+EA- years. In contrast, during NAO- and NAO-EA+ a more meridional distribution can be observed at the surface and offshore, although the NE–SW temperature gradient in deeper levels is persistent regardless of the climate phases.

The study suggests that stronger upwelling along the CCUS is observed during coupled NAO+EA- phases represented by maximum values of both Upwelling Indices and a deepening of the coastal ILD. It therefore emphasizes the impact of coupled phases of climate pattern on extreme events in the ocean. It is, however, necessary consider that the used dataset consists of modelled data. Even though the results are validated with in-situ measurements and satellite data, the model might not reflect the real conditions of the ocean structure.